# Autoregressive Video Generation with Learnable Memory and Consistent Decoding

## Abstract

Long-form video generation presents a dual challenge: models must capture long-range dependencies while preventing the error accumulation inherent in autoregressive decoding. To address these challenges, we make two contributions. First, for dynamic context modeling, we propose MemoryPack, a learnable context-retrieval mechanism that leverages both textual and image information as global guidance to jointly model short- and long-term dependencies, achieving minute-level temporal consistency. This design scales gracefully with video length, preserves computational efficiency, and maintains linear complexity. Second, to mitigate error accumulation, we introduce Direct Forcing, an efficient single-step approximating strategy that improves training–inference alignment and thereby curtails error propagation during inference. Together, MemoryPack and Direct Forcing substantially enhance the context consistency and reliability of long-form video generation, advancing the practical usability of autoregressive video models. Project website: `https://anonymous.4open.science/w/ICLR2026-55FF`.

## 1 Introduction

Video generation has emerged as a central problem in generative modeling, enabling applications in content creation (Chen et al., 2025), embodied intelligence (Wu et al., 2023; Liu et al., 2024; Cheang et al., 2024; Wu et al., 2024), and interactive gaming (Team, 2025; Li et al., 2025a). Recent Diffusion Transformer (DiT) models (Kong et al., 2024; Wan et al., 2025; Gao et al., 2025; Zhang et al., 2025b) demonstrate strong capabilities in capturing complex spatiotemporal dependencies and character interactions within fixed-length sequences, producing realistic video clips. However, long-form video generation remains challenging: the substantially larger token count of videos renders end-to-end modeling with quadratic-complexity DiT architectures computationally prohibitive, and the lack of effective long-term modeling leads to increasingly severe drift as video length grows. These factors pose significant challenges for generating minute-scale or longer videos while maintaining temporal coherence and computational efficiency.

Existing approaches (Teng et al., 2025; Zhang & Agrawala, 2025) typically enhance context consistency by retaining only the most recent frames or applying fixed compression strategies to select key frames. However, due to the limited window size and aggressive token compression, these rigid mechanisms rely primarily on local visual information and fail to capture global dependencies. In long-form video generation, the absence of global context inevitably degrades temporal coherence.

We address this challenge by reformulating long video generation as a long-short term information retrieval problem, where the model must effectively retrieve both persistent long-term context and dynamic short-term cues to guide frame synthesis. Specifically, we introduce MemoryPack, a linear-complexity dynamic memory mechanism that leverages textual and image information as global guidance. MemoryPack retrieves long-term video context that is semantically aligned with the overall narrative to reinforce temporal coherence, while simultaneously exploiting adjacent frames as short-term cues to enhance motion and pose fidelity. In contrast to methods based on fixed compression or frame selection, MemoryPack enables flexible associations between historical information and future frame generation.

Another central challenge in long-form video generation is error accumulation caused by the training–inference mismatch: during training, models condition on ground-truth frames, whereas

at inference they rely on self-predictions, causing errors to compound over long horizons. Although Zhang & Agrawala (2025) attempts to mitigate drift by generating frames in reverse order, the reliance on tail frames substantially reduces dynamic degree in long videos, limiting its effectiveness. Huang et al. (2025) addresses this by conditioning directly on generated outputs, but to improve efficiency it requires distribution distillation, which introduces extra computation and, due to the inherent limitations of distillation, the generation quality is degraded, and consequently, incorporating previously generated results into the training process introduces additional noise.

To address this issue, we introduce Direct Forcing, an efficient strategy that aligns training with the model's inference in a single step. Building on rectified flow (Liu et al., 2022), we perform one-step backward ODE computation in the predicted vector field to approximate inference outputs. This method incurs no additional overhead, requires no distillation, preserves train–inference consistency, and mitigates error accumulation.

Our method achieves state-of-the-art performance on VBench (Huang et al., 2024a) across key metrics, including Motion Smoothness, Background Consistency, and Subject Consistency, while further enhancing robustness against error accumulation. Experimental results demonstrate that MemoryPack and Direct Forcing effectively model long-term contextual information and achieve superior consistency performance.

In summary, our contributions are threefold:

- **MemoryPack**: a dynamic memory mechanism that leverages text and image as global guidance to retrieve historical context, while exploiting adjacent frames as short-term cues. This design enables efficient modeling of minute-level temporal consistency without relying on rigid compression.

- **Direct Forcing**: a single-step approximating strategy that aligns training with inference efficiently, eliminating distillation and mitigating error accumulation in long-horizon generation.

- **State-of-the-Art Performance**: extensive evaluations demonstrate that our approach achieves state-of-the-art results in long-term consistency metrics.

## 2 RELATED WORKS

### 2.1 EFFICIENT ACCUMULATE FOR LONG VIDEO GENERATION

Long video generation remains an open problem. Since most existing approaches are still built upon the DiT framework, they suffer from the $\mathcal{O}(L^2)$ computational complexity, where $L$ denotes the number of context tokens, which becomes prohibitively large for long sequences. To address this issue, several strategies have been proposed. Some works replace the original mechanism with linear attention (Cai et al., 2022; Gu & Dao, 2023b; Yang et al., 2024a; Wang et al., 2020; Peng et al., 2023; Gu & Dao, 2023a) to improve efficiency. Dalal et al. (2025) alleviates training costs through test-time training, while Huang et al. (2025) accelerates training by employing a KV-cache for rolling updates. In addition, a training-free approach, Xi et al. (2025) (SVG), exploits the inherent sparsity of 3D full attention to further enhance inference efficiency.

### 2.2 FRAMEWORK FOR VIDEO GENERATION

Current video generation research (Zhang et al., 2025a; Gao et al., 2025; Zhang et al., 2025b; Wan et al., 2025; Kong et al., 2024; Yang et al., 2024b; Zheng et al., 2024; Peng et al., 2025) has primarily focused on short clips at the second level. Recently, several studies have explored extending such models to generate minute-long videos. For example, Zhang & Agrawala (2025) and Teng et al. (2025) adopt DiT as the backbone network and iteratively predict future frames conditioned on previously generated ones. However, these approaches face challenges in modeling long-term dependencies. In particular, they restrict the model to attend only to the most recent frames, which improves computational efficiency but inevitably leads to the loss of long-range information. In contrast, Cai et al. (2025) introduces the Mixture of Contexts (MoC), which selects contextual information through routing. While this design improves flexibility in leveraging context, it relies on

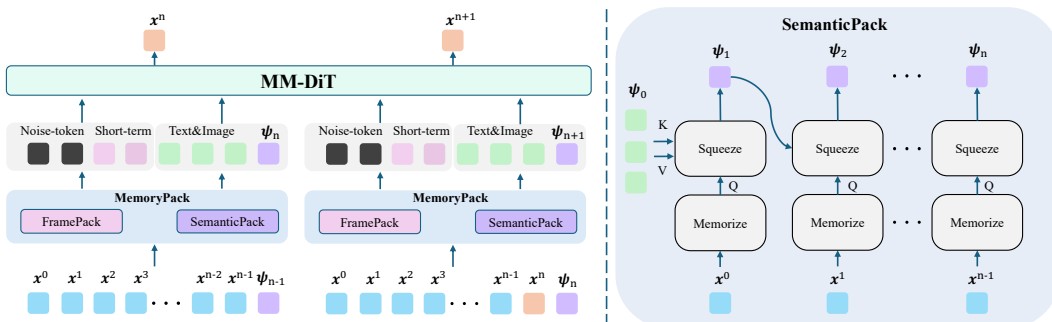

Figure 1: **Overview of our framework.** Given a text prompt, an input image, and history frames, the model autoregressively generates future frames. Prior to feeding data into MM-DiT, MemoryPack retrieves both long- and short-term context. In SemanticPack, visual features are extracted within local windows via self-attention, followed by cross-attention to align them with global textual and visual information. This design achieves linear computational complexity and substantially improves the efficiency of long-form video generation.

manually defined selection rules, thereby limiting the model's ability to autonomously determine relevant information.

## 3 METHOD

Given $n$ historical segments $\{\mathbf{x}^0, \ldots, \mathbf{x}^{n-1}\}$, along with a textual prompt $P$ and a conditional image $I$, our objective is to generate the subsequent segment $\mathbf{x}^n$. We formulate this as an autoregressive image-to-video generation task, enabling the synthesis of arbitrarily long video sequences from both textual and image inputs. Unless otherwise specified, all references to segments in this paper denote their latent-space representations. Our approach builds upon the Diffusion Transformer (DiT) architecture (Peebles & Xie, 2023; Kong et al., 2024) to create an autoregressive model for future segments generation, with its overall architecture illustrated in Figure 1. To enhance temporal consistency and mitigate error accumulation—inherent challenges in autoregressive generation—we introduce two key innovations: (i) MemoryPack, a hierarchical and efficient fusion module that leverages the text prompt and image as a global guide to model both long- and short-term temporal dependencies (Sec. 3.1); and (ii) the Direct Forcing strategy, which aligns the training process with inference by employing single-step approximation. This approach mitigates the discrepancy between conditioning on ground-truth segments during training and on model-generated segments during inference (Sec. 3.2).

### 3.1 MEMORYPACK

Generating long-form videos requires balancing high-fidelity, smooth local motion with semantic coherence across the global narrative. Previous works (Teng et al., 2025; Lin et al., 2025) typically condition video generation on a fixed sliding window of recent segments $\{\mathbf{x}^{n-k}, \ldots, \mathbf{x}^{n-1}\}$ together with a text prompt $P$ and a conditional image $I$. While such methods excel at preserving local motion, they often fail to capture long-range dependencies, including object identities and scene layouts. Conversely, models focusing solely on distant context may lose track of fine-grained motion cues, leading to artifacts and temporal discontinuities.

To address these challenges, we introduce MemoryPack, a hierarchical module that jointly leverages complementary short-term and long-term contexts for video generation. It consists of two components: FramePack (Zhang & Agrawala, 2025) and SemanticPack.

FramePack focuses on short-term context by capturing appearance and motion through a fixed compression scheme, thereby enforcing short-term consistency. However, its fixed window size and compression ratio constrain its ability to dynamically propagate information over long time horizons.

To maintain global temporal coherence, SemanticPack integrates visual features with textual and image guidance, unlike prior methods (Cai et al., 2025) that rely solely on visual representations. This is achieved by iteratively updating a long-term memory representation $\psi$ using contextual video segments $\{\mathbf{x}^0, \ldots, \mathbf{x}^{n-1}\}$, a text prompt $P$, and a reference image $I$. The process consists of two structured operations: (1) Memorize, which applies self-attention within windows of historical segments to produce compact embeddings. This approach mitigates the prohibitive quadratic complexity of attending to long histories while retaining holistic window-level cues. (2) Squeeze, which then injects the textual and image guidance into this visual memory. Following prior work (Wan et al., 2025), we implement this as a cross-attention layer where the output of Memorize serves as the query, and the representation $\psi$ acts as the key and value. This alignment ensures the long-term memory remains globally aware and semantically grounded:

$$\psi_{n+1} = \text{Squeeze}\big(\psi_n, \text{Memorize}(\mathbf{x}^n)\big). \tag{1}$$

For initialization ($n = 0$), we set $\psi_0$ as the concatenation of the prompt feature and the reference image feature, providing a semantic prior that anchors the memory trajectory. Importantly, the computational complexity of SemanticPack is $\mathcal{O}(n)$, ensuring scalability by preventing costs from growing prohibitively with the number of historical frames. By integrating both short-(FramePack) and long-term(SemanticPack) context, MemoryPack retains information from distant frames while preserving temporal and motion consistency with high computational efficiency. Additional experiments on SemanticPack are provided in the appendix.

**RoPE Consistency**    In DiT-based autoregressive video generation, long videos are typically partitioned into multiple segments during training. However, this segmentation causes even adjacent segments from the same video to be modeled independently, leading to the loss of cross-segment positional information and resulting in flickering or temporal discontinuities. To address this issue, we treat the input image as a CLS-like token and incorporate RoPE (Su et al., 2024) to explicitly encode relative positions across segments. Specifically, during training, for each video clip, we assign the image the initial index of the entire video, thereby preserving coherence and enhancing global temporal consistency. Formally, RoPE satisfies

$$R_q(\boldsymbol{x}_q, m)R_k(\boldsymbol{x}_k, n) = R_g(\boldsymbol{x}_q, \boldsymbol{x}_k, n-m), \quad \Theta_k(\boldsymbol{x}_k, n) - \Theta_q(\boldsymbol{x}_q, m) = \Theta_g(\boldsymbol{x}_q, \boldsymbol{x}_k, n-m), \tag{2}$$

where $R_q, R_k, R_g$ denote the rotation matrices for query, key, and relative position, respectively; $\Theta_q, \Theta_k, \Theta_g$ denote the corresponding rotation angles. $\boldsymbol{x}_q$ and $\boldsymbol{x}_k$ are the query and key vectors, with $m$ and $n$ being their position indices.By assigning the image token an index of start, the sequence can jointly capture absolute positions across video segments and relative dependencies within each segment, thereby mitigating flickering and discontinuities.

## 3.2 Direct Forcing

Autoregressive video generation often suffers from error accumulation caused by the discrepancy between training and inference: during training, the model is conditioned on ground-truth frames, whereas during inference it relies on its own previously generated outputs.

To mitigate this mismatch, we propose a rectified-flow-based single-step approximation strategy that directly aligns training and inference trajectories while preserving computational efficiency.

**Training with Rectified Flow.**    Following the rectified flow formulation Liu et al. (2022), we define a linear interpolation between the video distribution $\mathbf{x}$ and Gaussian noise $\boldsymbol{\epsilon} \sim \mathcal{N}(0, I)$. For simplicity, we omit the superscript of $\mathbf{x}$ and use the subscript to indicate time $t$:

$$\mathbf{x}_t = t\mathbf{x} + (1-t)\boldsymbol{\epsilon}, \quad t \in [0, 1]. \tag{3}$$

The instantaneous velocity along this trajectory is given by

$$\boldsymbol{u}_t = \frac{d\mathbf{x}_t}{dt} = \mathbf{x} - \boldsymbol{\epsilon}, \tag{4}$$

which defines an ordinary differential equation (ODE) guiding $\mathbf{x}_t$ toward the target $\mathbf{x}$. The model predicts a velocity field $v_\theta(\mathbf{x}_t, t)$, and parameters are optimized by minimizing the flow matching loss:

$$\mathcal{L}_{\text{FM}}(\theta) = \mathbb{E}_{t, \mathbf{x}, \boldsymbol{\epsilon}}\big[\|v_\theta(\mathbf{x}_t, t) - \boldsymbol{u}_t\|^2\big]. \tag{5}$$

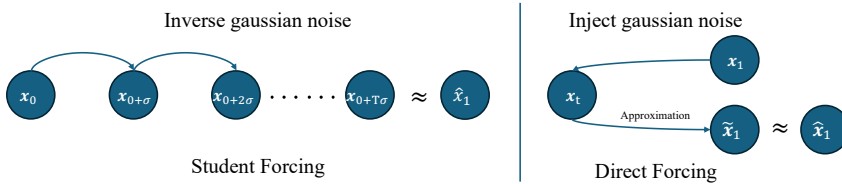

Figure 2: Schematic illustration of the approximation process. In Student Forcing, multi-step inference is applied to approximate $\hat{\mathbf{x}}_1$, but this incurs substantial computational overhead and slows training convergence. In contrast, Direct Forcing applies a single-step transformation from $\mathbf{x}_1$ to $\mathbf{x}_t$, followed by a denoising step that produces $\tilde{\mathbf{x}}_1$ as an estimate of $\hat{\mathbf{x}}_1$. This approach incurs no additional computational burden, thereby enabling faster training.

**Single-Step Approximation.** During inference, a video is generated by reverse-time integration of the ODE starting from $\mathbf{x}_0 \sim \mathcal{N}(0, I)$ to generated $\hat{\mathbf{x}}_1$:

$$\hat{\mathbf{x}}_1 = \int_0^1 v_\theta(\mathbf{x}_t, t)dt. \tag{6}$$

As illustrated in Fig. 2, Student Forcing (Bengio et al., 2015), through multi-step inference in Eq. 6 as an approximation of $\hat{\mathbf{x}}_1$, would incur significant computational costs. Based on Eq. 7, we align training with inference without incurring high computational costs by approximating the trajectory in a single step:

$$\tilde{\mathbf{x}}_1 = \mathbf{x}_t + \Delta_t * v_\theta(\mathbf{x}_t, t) \approx \hat{\mathbf{x}}_1, \quad \Delta_t = 1 - t \tag{7}$$

Intuitively, since rectified flow guarantees a more direct ODE trajectory, $\mathbf{x}_t + \Delta_t v_\theta(\mathbf{x}_t, t)$ serves as an effective single-step approximation of the generated distribution, thereby bridging the gap between training and inference. Concretely, the model first uses the ground-truth data $\mathbf{x}^{i-1}$ and Eq. 7 to obtain a one-step approximation $\tilde{\mathbf{x}}^{i-1}$. This approximation then serves as the conditional input for generating $\mathbf{x}^i$ during training, thereby exposing the model to inference-like conditions and effectively mitigating distribution mismatch while reducing error accumulation.

**Optimization Strategy** In practice, we leverage Direct Forcing to sample clips from the same video in chronological order and use them as conditional inputs for iterative training. Unlike prior approaches (Dalal et al., 2025; Zhao et al., 2024) that train clips independently, this design explicitly reinforces temporal continuity across segments. However, applying backpropagation with an optimizer update at every step can perturb the learned distribution and impair the consistency of input distributions across clips during training. To mitigate this issue, we adopt gradient accumulation: gradients are aggregated over multiple clips before performing a single parameter update. This strategy stabilizes optimization and, as our experiments show, substantially improves cross-clip consistency and long-range temporal coherence in generated videos.

## 4 EXPERIMENT

### 4.1 IMPLEMENTATION DETAILS

We adopt Framepack-F1 as the backbone model for the image-to-video generation task. The training dataset is primarily sourced from Mira (Ju et al., 2024) and Sekai (Li et al., 2025b), comprising approximately 16,000 video clips with a total duration of 150 hours across diverse scenarios. The longest videos in both Mira and Sekai extend up to one minute. To ensure data quality, we apply dynamism and shot-cut filtering to all samples. Training is conducted in parallel on GPU clusters (96GB memory each) with a batch size of 1 for approximately five days. We employ the AdamW optimizer with an initial learning rate of $10^{-5}$. The training procedure consists of two stages. In the first stage, we apply teacher forcing to train the entire network, which accelerates convergence and mitigates instability caused by sampling bias. In the second stage, we only fine-tune the output layer with Direct Forcing, which could stabilize the backbone and align training with inference, thereby substantially reducing error accumulation in autoregressive generation.

| Method | Global Metrics | | | | | | Error Accumulation | | | | Human Evaluation | |
|---|---|---|---|---|---|---|---|---|---|---|---|---|
| | Imaging Quality ↑ | Aesthetic Quality ↑ | Dynamic Degree ↑ | Motion Smoothness ↑ | Background Consistency ↑ | Subject Consistency ↑ | ΔImaging Quality ↓ | ΔAesthetic Quality ↓ | ΔBackground Consistency ↓ | ΔSubject Consistency ↓ | ELO ↑ | Rank ↓ |
| Magi-1 | 54.64% | 53.98% | **67.5%** | 99.17% | 89.30% | 82.33% | 4.19% | 5.02% | 1.53% | 0.97% | 1434 | 4 |
| FramePack-F0 | **68.06%** | **62.89%** | 15.38% | 99.27% | 92.22% | 90.03% | **2.34%** | **2.45%** | 2.68% | 2.99% | 1459 | 3 |
| FramePack-F1 | 67.06% | 59.11% | 53.85% | 99.12% | 90.21% | 83.48% | 2.71% | 4.57% | 1.59% | 1.08% | 1537 | 2 |
| Ours | 67.55% | 59.75% | 48.37% | **99.31%** | 93.25% | 91.16% | 2.51% | 3.25% | **1.21%** | 0.76% | **1568** | **1** |

Table 1: Comparison of different methods on **Global Metrics**, **Error Accumulation Metrics**, and **Human Evaluation**. The best results are highlighted in **bold**.

## 4.2 EVALUATION METRICS

Following established evaluation protocols (Huang et al., 2024a; Zheng et al., 2025), we assess the generated videos using six quantitative metrics: (1) imaging quality, (2) aesthetic quality, (3) dynamic degree, (4) motion smoothness, (5) background consistency, and (6) subject consistency. In addition, we conduct exposure bias and human evaluation to provide complementary subjective assessments. To distinguish the model's generation capability across different temporal scales, we categorize videos into three duration ranges: short (10 seconds), medium-length (30 seconds), and long (1 minute).

**Imaging Quality:** This metric captures distortions in generated frames, including over-exposure, noise, and blur. We measure it using the MUSIQ (Ke et al., 2021) image quality predictor trained on the SPAQ (Fang et al., 2020) dataset.

**Aesthetic Quality:** We evaluate the aesthetic quality of each video frame using the LAION aesthetic predictor (Schuhmann et al., 2022), which considers factors such as composition, color richness and harmony, photorealism, naturalness, and overall artistic value.

**Dynamic Degree:** We employ RAFT (Teed & Deng, 2020) to estimate the extent of motion in synthesized videos.

**Motion Smoothness:** We leverage motion priors from the video frame interpolation model (Li et al., 2023), as adapted by VBench, to evaluate motion smoothness.

**Background Consistency:** Following VBench, we measure the temporal consistency of background scenes by computing CLIP (Radford et al., 2021) feature similarity across frames.

**Subject Consistency:** We compute DINO (Caron et al., 2021) feature similarity between frames to assess the consistency of a subject's appearance throughout the sequence.

**Exposure Bias Metric:** Following Zhang & Agrawala (2025), we evaluate long-horizon video generation by defining the start–end contrast, denoted as $\Delta_{\text{drift}}^{M}$, for an arbitrary quality metric $M$ (e.g., imaging quality, aesthetic quality).

$$\Delta_{\text{drift}}^{M}(V) = \big|M(V_{\text{start}}) - M(V_{\text{end}})\big|, \tag{8}$$

Where V is the tested video, $V_{\text{start}}$ t represents the first 15% of frames, and $V_{end}$ represents the last 15% of frames. To accurately measure the model's long-horizon generation capability, we take the last frame generated by the model itself as $V_{\text{end}}$, rather than the final frame of the user-obtained video. Our rationale is that the evaluation should focus on the model's capability, and thus extrapolation ought to be defined with respect to the model's own outputs.

**Human Assessment:** We collect human preferences through A/B testing. Specifically, we generate 160 videos, randomly shuffle their order, and distribute them to evaluators to ensure unbiased assessments. Following Zhang & Agrawala (2025), we report ELO-K32 scores along with the corresponding rankings.

## 4.3 GENERATION PERFORMANCE

We compare our method with FramePack-F0, FramePack-F1 (Zhang & Agrawala, 2025), and Magi-1 (Teng et al., 2025). Evaluation is conducted on 160 videos: 60 of 10 seconds, 60 of 30 seconds, and 40 of 60 seconds, all at 480p and 24 fps. All images are sourced from Vbench (Huang et al., 2024b), and prompts are rewritten using Qwen2.5-Vl (Bai et al., 2025). Quantitative results are reported in Table 1 and Table 2, while qualitative comparisons are shown in Fig. 3, Fig. 4, and Fig. 5.

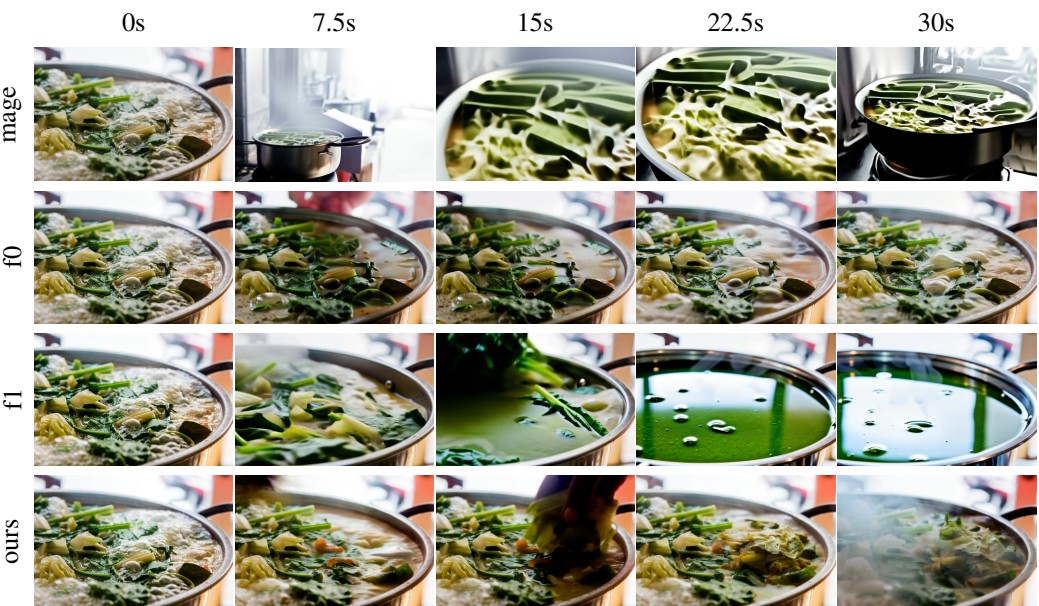

Figure 3: Visualization of 30-second videos comparing all methods in terms of consistency preservation and interaction capability. **Prompt:** Close-up view of vegetables being added into a large silver pot of simmering broth, with leafy greens and stems swirling vividly in the bubbling liquid. Rising steam conveys warmth and motion, while blurred kitchen elements and natural light in the background create a homely yet dynamic culinary atmosphere.

**Qualitative results.** Representative 30-second and 60-second examples are shown in Fig. 3 and Fig. 4. For brevity, we simplify the prompts associated with generated videos. Our method exhibits fewer temporal identity shifts and geometric distortions compared to FramePack-F1 and Magi-1. While FramePack-F0 preserves visual fidelity in Fig. 3, it demonstrates reduced inter-frame dynamics, consistent with its quantitative metrics. Notably, as video length increases, competing methods suffer from more severe error accumulation, whereas our approach maintains video quality, including aesthetics and character consistency. These results highlight the effectiveness of Direct Forcing in enhancing long-term video generation.

**Quantitative results.** As reported in Table 1, our method achieves the best performance on Background Consistency, Subject Consistency, and Motion Smoothness, demonstrating a strong ability to preserve long-term temporal coherence and generate smooth motion. While FramePack-F0 attains higher Image Quality and Aesthetic Quality, it exhibits weaker inter-frame dynamics, which we attribute to its anti-drift sampling strategy. Conversely, Magi-1 produces stronger dynamics but suffers from degraded temporal and subject consistency. In human evaluation, our method with ELO-32K still achieves the best overall performance. These findings are further corroborated by the qualitative results in Fig. 3 and Fig. 4. Importantly, our approach also achieves the lowest Error Accumulation, underscoring its stability in long-term video generation.

**Ablation study.** As reported in Table 2, all experimental results were obtained using a subset of the training data to conserve computational resources. To assess the influence of the training dataset, we first fine-tuned FramePack-F1 (F1 with CT) on our dataset. This slightly improved the Dynamic Degree metric but degraded performance on other evaluations, demonstrating the effectiveness of our proposed MemoryPack and Direct Forcing.

To further examine the semantic contribution of MemoryPack, we initialized the global memory $\psi_0$ with a zero vector (zero-MemoryPack). This led to worse performance on error-accumulation metrics, which we attribute to the absence of semantic guidance, resulting in reduced consistency. These findings indicate that semantic guidance stabilizes long-term video generation.

| Method | Global Metrics | | | | | | Error Accumulation | | | |
|---|---|---|---|---|---|---|---|---|---|---|
| | Imaging Quality ↑ | Aesthetic Quality ↑ | Dynamic Degree ↑ | Motion Smoothness ↑ | Background Consistency ↑ | Subject Consistency ↑ | ΔImaging Quality ↓ | ΔAesthetic Quality ↓ | ΔBackground Consistency ↓ | ΔSubject Consistency ↓ |
| F1 w/ CT | 54.33% | 53.07% | 56.77% | 98.81% | 85.32% | 85.64% | 4.32% | 7.18% | 3.22% | 1.32% |
| MemoryPack | 55.31% | **60.69%** | 51.13% | 98.86% | **88.21%** | 86.77% | **2.47%** | 4.99% | 2.31% | 1.88% |
| zero-MemoryPack | 57.37% | 51.91% | **62.71%** | 98.85% | 87.31% | 86.21% | 8.32% | 6.31% | 2.43% | 3.10% |
| Student forcing | 49.32% | 52.33% | 59.88% | 98.96% | 85.32% | 82.41% | 3.17% | 5.11% | **2.02%** | 1.12% |
| All | **57.65%** | 55.75% | 55.37% | **99.11%** | 87.17% | **88.77%** | 3.21% | **4.77%** | **2.02%** | **0.99%** |

Table 2: Ablation Study on Different Model Structures and Strategies.

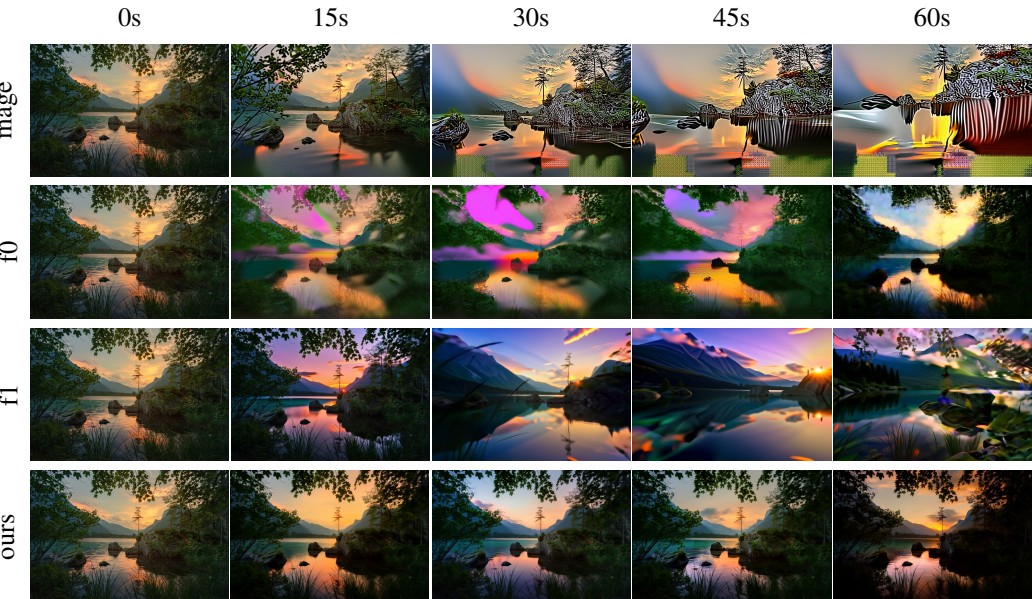

Figure 4: Visualization of a 60-second video illustrating the accumulation of errors. Our method maintains image quality comparable to the first frame even over minute-long sequences. **Prompt:** The sun sets over a serene lake nestled within majestic mountains, casting a warm, golden glow that softens at the horizon. The sky is a vibrant canvas of orange, pink, and purple, with wispy clouds catching the last light. Calm and reflective, the lake's surface mirrors the breathtaking colors of the sky in a symphony of light and shadow. In the foreground, lush greenery and rugged rocks frame the tranquil scene, adding a sense of life and stillness. Majestic, misty mountains rise in the background, creating an overall atmosphere of profound peace and tranquility.

We also ablated Direct Forcing by training the model with its actual sampling process as input. To balance training efficiency, we set the sampling step to 5; however, the resulting performance remained substantially inferior to Direct Forcing, underscoring its effectiveness for long-term video generation.

**Consistency visualization.** Figure 5 illustrates object reconstruction after prolonged disappearance. Over a 60-second sequence, our method accurately reconstructs objects that remain absent for extended periods. Notably, even when subjects temporarily vanish due to occlusion, the model reconstructs and generates objects with consistent identity and 2D structure after long intervals. These results demonstrate that MemoryPack effectively preserves long-term contextual information, enabling stable memory for extended video generation.

## 5 CONCLUSION

We propose MemoryPack, a lightweight mechanism that jointly models long- and short-term memory from visual inputs and text prompts, enabling the learning of long-range dependencies without requiring 3D priors, handcrafted heuristics, or modifications to the core DiT framework. We further introduce Direct Forcing, a simple yet effective approach for mitigating the training–inference

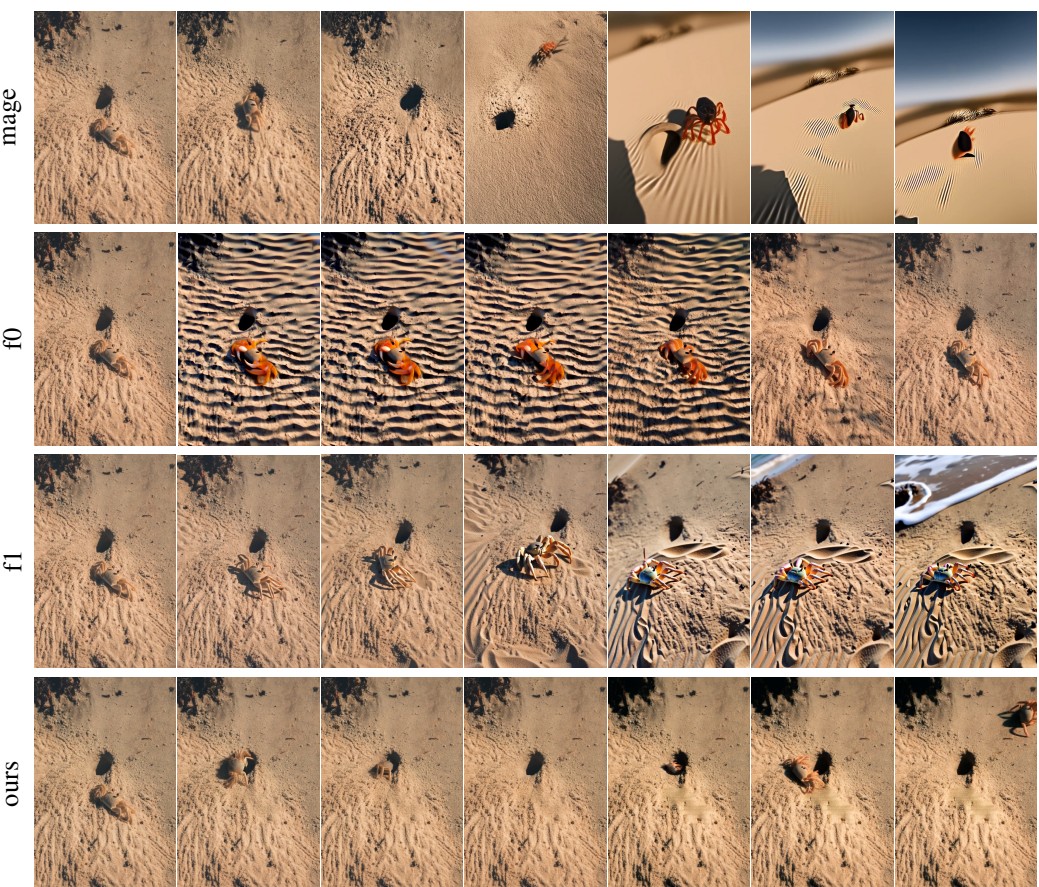

Figure 5: Consistency evaluation on a 60-second video shows that when an object ID is heavily occluded for an extended period, reconstruction remains challenging. Both F0 and F1 fail to follow the prompt and exhibit noticeable error accumulation. Although MAGI-1 follows the prompt, it is unable to maintain temporal consistency. **Prompt:** On the peaceful, sun-drenched sandy beach, a small crab first retreats into its burrow before reemerging. The lens captures its shimmering shell and discreet stride under the low sun angle. As it slowly crawls outward, the crab leaves a faint trail behind, while its elongated shadow adds a cinematic texture to this tranquil scene.

discrepancy in autoregressive video generation, which incurs no additional training cost or computational overhead. Experimental results show that MemoryPack and Direct Forcing reduce error accumulation, enhance identity-preservation consistency, and alleviate exposure bias. Together, these contributions pave the way for next-generation long-form video generation models.

**Limitation :** This work explores MemoryPack for modeling both long- and short-term memory. In contrast, SemanticPack employs a sliding-window Transformer backbone to capture long-term information. This design achieves linear computational complexity and enables efficient training and inference. However, it still introduces artifacts in highly dynamic scenarios, and its ability to maintain long-term consistency in hour-long videos remains limited. Similarly, Direct Forcing adopts a simple and efficient single-step approximation strategy, yet its effectiveness is highly dependent on the pre-trained model. As a result, training currently requires multiple stages, and whether this fitting strategy can be integrated into a single-stage pipeline remains an open question. We leave these directions for future work.

## 6 ETHICS STATEMENT

This work focuses on algorithmic advances in long-form video generation. Our study does not involve sensitive personal data, or copyrighted materials. The datasets used (e.g., Li et al. (2025b); Ju et al. (2024)) are publicly available benchmarks that comply with community standards. our research is centered on methodological improvements in temporal coherence and computational efficiency. We encourage responsible use of the proposed techniques and recommend safeguards such as watermarking and content moderation in downstream applications.

## 7 REPRODUCIBILITY STATEMENT

We have made substantial efforts to ensure the reproducibility of our work. Detailed descriptions of the proposed MemoryPack and Direct Forcing mechanisms are provided in Sections 3.1 and 3.2, respectively. To further support reproducibility, we will release source code, configuration files, and pretrained checkpoints upon publication. These resources will enable researchers to replicate our experiments under the same settings and adapt the framework to new datasets or tasks with minimal modifications. Comprehensive documentation and usage guidelines will also be provided to lower the barrier to reproduction and extension. We hope these efforts will not only validate our findings but also foster future research in long-term video generation.

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

# A APPENDIX

We provide additional experimental details and visualizations in this section. In Sec. A.1, we present the optional structures of SemanticPack, where we conduct both qualitative and quantitative evaluations to analyze their impact on modeling efficiency, representation capacity, and overall consistency performance. Sec. A.2 describes the training setup of Direct Forcing, including implementation details, hyperparameter configurations, and training dynamics, to facilitate reproducibility and provide deeper insight into its effectiveness. In Sec. A.3, we provide consistency visualizations, including minute-level examples, to evaluate the model's ability to reconstruct historical information under long-term sequences and severe occlusions, while preserving spatial layouts, maintaining subject fidelity, and mitigating temporal drifting.

## A.1 VALIDATION OF SEMANTICPACK MODELS

We conducted further ablation studies against Squeeze to evaluate the network's capacity for capturing long-term dependencies. As shown in Fig. 6, we designed three fusion schemes: (a) text and image features are used as activation vectors for $K$ and $V$, while the visual representation serves as the query; (b) text and image features are used as the query, with the visual representation as key/value; (c) to enrich the query, we concatenate text and image features with the visual representation from the first window, and then follow the same setting as (b) for subsequent steps.

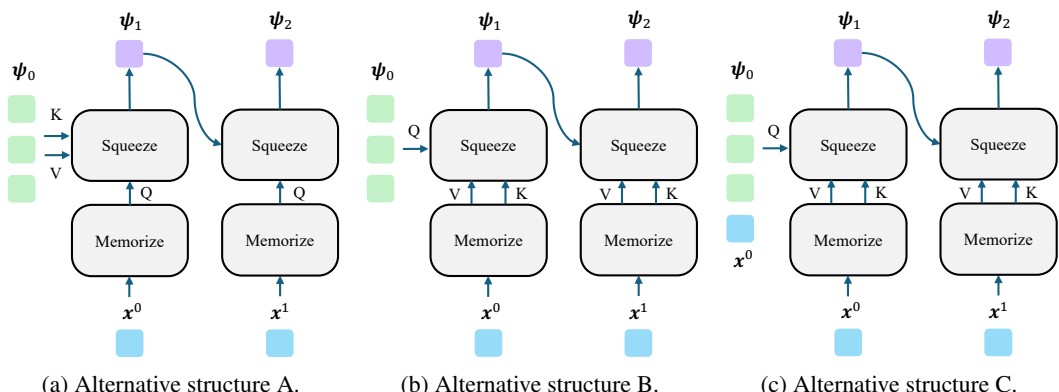

(a) Alternative structure A.    (b) Alternative structure B.    (c) Alternative structure C.

Figure 6: Illustration of the optional architecture of SemanticPack.

In Tab. 3, we present quantitative results. We observe that structures B and C cause a notable degradation in visual quality and a substantial reduction in temporal dynamics. We attribute these effects to the following reasons: in structure B, the number of text and image tokens is considerably smaller than the number of query tokens in structure A, which is insufficient to capture adequate visual representations, thereby impairing the model's ability to model dynamics. In structure C, although incorporating the initial windows into the query increases the token count, it also introduces multi-modal information, thereby increasing training difficulty.

| Method | Global Metrics | | | | | |
| --- | --- | --- | --- | --- | --- | --- |
| | Imaging Quality ↑ | Aesthetic Quality ↑ | Dynamic Degree ↑ | Motion Smoothness ↑ | Background Consistency ↑ | Subject Consistency ↑ |
| C | 48.31% | 53.15% | 28.98% | 97.72% | 83.27% | 83.74% |
| B | 50.11% | 50.71% | 32.78% | **98.91%** | 87.11% | 80.56% |
| A | **55.31%** | **60.69%** | **51.13%** | 98.86% | **88.21%** | **86.77%** |

Table 3: Ablation Study on Different Model Structures and Strategies. Among all variants, structure A achieves the best overall performance.

To further validate our conclusions, we conduct visualization experiments for the three structures presented in Fig. 7. The results reveal clear differences in temporal consistency: while structure A

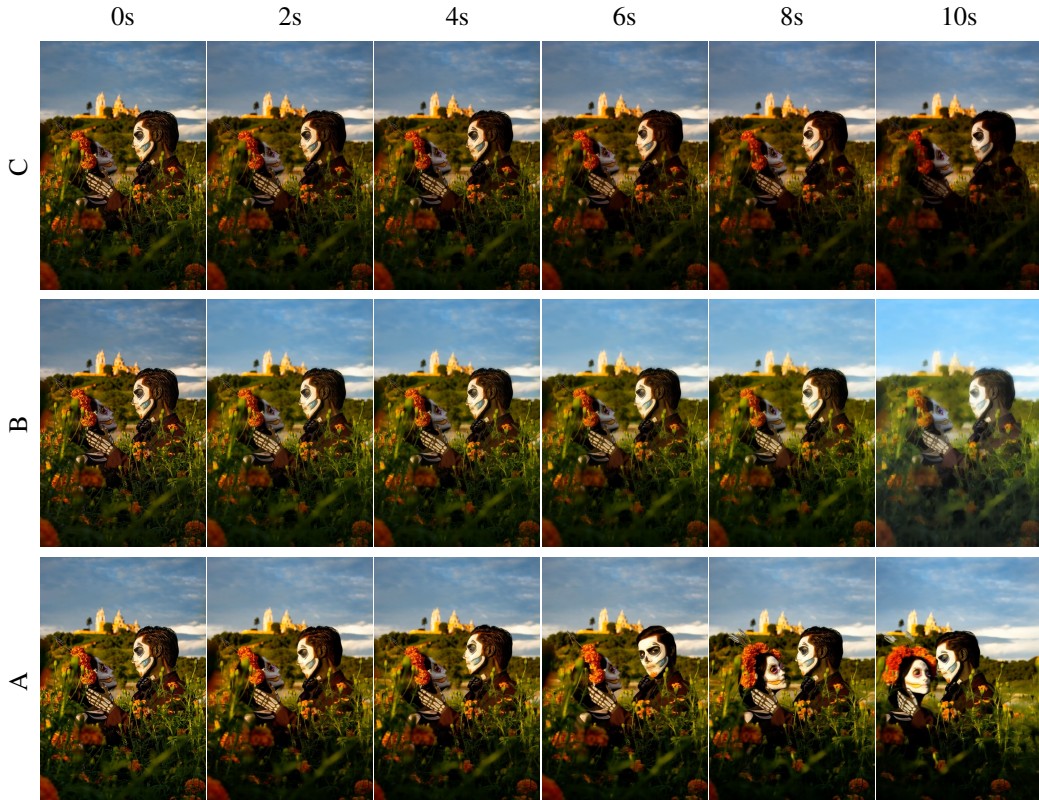

Figure 7: **Prompt:** In a field of golden marigolds, a man and woman stood entwined, their faces glowing with intricate sugar skull makeup beneath the setting sun. The woman, crowned with fiery orange blossoms, gazed at him with tender devotion. He met her eyes, the bold black-and-white patterns on his face striking against his chestnut jacket, his hands gently interlaced with hers. Turning briefly toward the camera, he then lowered his head to kiss her. Behind them, hilltops crowned with golden-domed buildings shimmered beneath a sky of soft blues and pinks, completing the serene, magical scene.

maintains relatively stable motion and coherent frame-to-frame transitions, structures B and C exhibit a pronounced degradation in temporal dynamics. In particular, both B and C suffer from noticeable blurred object boundaries over time, indicating their limited capacity to preserve long-range dependencies. Moreover, they accumulate substantially higher exposure bias compared to structure A, which further amplifies temporal drift and undermines overall video fidelity.

## A.2 DIRECT FORCING TRAINING DETAILS

For Direct Forcing, we employ a data sampler to standardize video clips and ensure sequential training. Since all clips are accumulated through gradient accumulation, the effective number of optimization steps becomes limited. To mitigate this issue, we adopt a curriculum learning strategy by sorting videos according to their length. Specifically, we begin training on videos containing a single clip, which effectively reduces the task to an image-to-video setting and lowers the training difficulty. We then progressively increase the number of accumulated clips, enabling the model to gradually align training with inference, while naturally extending to the autoregressive generation paradigm. This staged progression effectively enhances training stability. In terms of trainable parameters, we restrict updates to the final normalization layers and the output linear layer. Our rationale is that the discrepancy between training and inference primarily arises from a small number of accumulated errors, which eventually lead to drift; hence, adapting only the final layers is sufficient to correct this mismatch.

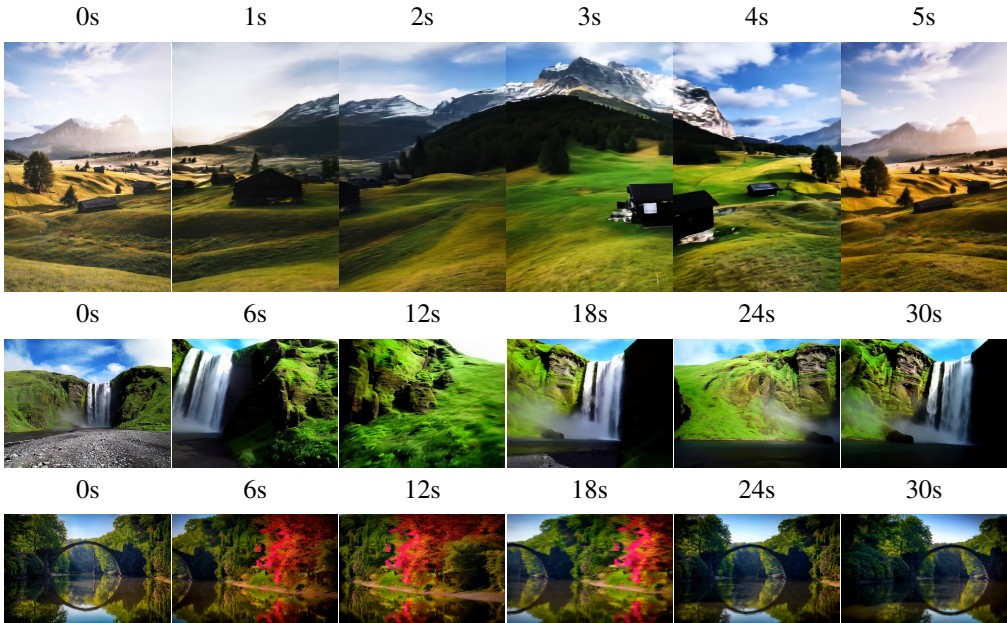

Figure 8: Visualization of long-term consistency. The model is evaluated on videos of 5 s and 30 s to assess its ability to maintain coherence.

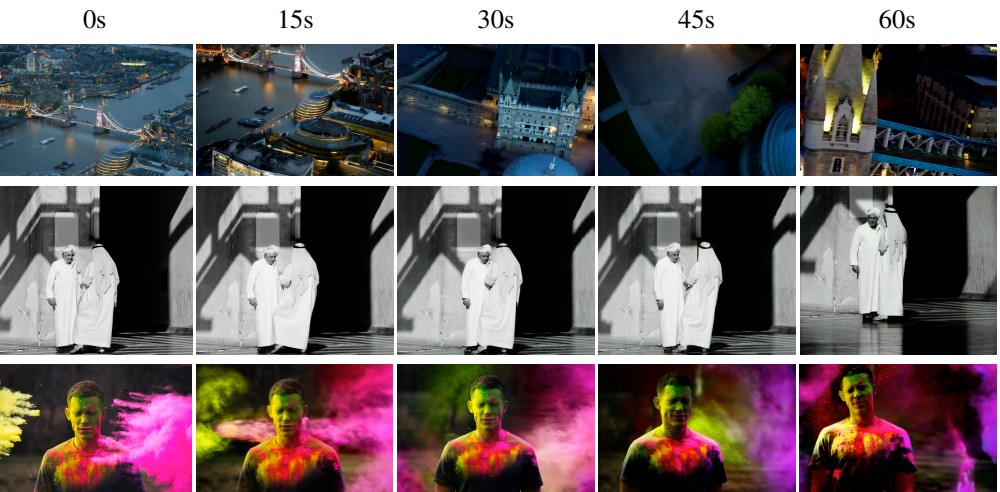

Figure 9: Visualization of 60s videos.

### A.3 VISULIZATION

We provide additional visualizations of consistency in Fig. 8. The results show that our model maintains long-term coherence across extended sequences and successfully recovers spatial layouts that are often disrupted by complex camera motion, such as panning and zooming. Furthermore, we present additional minute-level visualizations in Fig. 9, which highlight the model's ability to preserve scene structure, maintain subject integrity, and avoid temporal drift over extended durations.

### A.4 THE USE OF LARGE LANGUAGE MODELS (LLMS)

This thesis employs large language models (LLMs) to polish the writing and correct grammatical errors.

