# OpenReview forum: "Autoregressive Video Generation with Learnable Memory and Consistent Decoding"
_ICLR.cc/2026/Conference — ICLR 2026 Conference Withdrawn Submission_

### Official Review · Reviewer_3UCs · 2025-10-15

**Soundness:** 2
**Presentation:** 2
**Contribution:** 1
**Rating:** 4
**Confidence:** 3

**Summary:**

The authors propose two strategies to improve the performance of autoregressive video generation: 1) Memorypack, which extends Framepack by additionally incorporating a learnable token, and 2) Direct Forcing, which aims to mitigate the train-test discrepancy by training the network with the network outputs, rather than only the data source.

**Strengths:**

1. From the results, it can be seen that the proposed method outperforms FramePack.

2. The design of MemoryPack is sound, and improves upon FramePack.

**Weaknesses:**

Direct forcing is a straightfoward method to mitigate train-test gap by leveraging its own outputs during training. This is standard not only in diffusion models [1], but also in other domains [2,3]. The only difference is that it is now applied to FM models, and this **does not** guarantee that better one-step approximations are made, unless reflow is applied. Reflow is not applied in this model. From this, the only real contribution that I can find is MemoryPack.


**References**

[1] Chen, Ting, Ruixiang ZHANG, and Geoffrey Hinton. "Analog Bits: Generating Discrete Data using Diffusion Models with Self-Conditioning." ICLR 2022.

[2] Bengio, Samy, et al. "Scheduled sampling for sequence prediction with recurrent neural networks." Advances in neural information processing systems 28 (2015).

[3] Lamb, Alex M., et al. "Professor forcing: A new algorithm for training recurrent networks." Advances in neural information processing systems 29 (2016).

**Questions:**

1. Does the proposed network extrapolate to longer videos that exceed much longer than a minute?

2. Can this be trained from scratch, without being initialized from FramePack?

---

### Official Review · Reviewer_Pazf · 2025-10-31

**Soundness:** 2
**Presentation:** 3
**Contribution:** 2
**Rating:** 4
**Confidence:** 4

**Summary:**

This paper tackles the challenge of long-form autoregressive video generation, where maintaining temporal consistency and mitigating error accumulation are major bottlenecks. The authors propose two core contributions: (1) MemoryPack, a learnable memory retrieval module that jointly models short- and long-term dependencies using both visual and textual cues, achieving linear complexity for scalable long video generation. (2) Direct Forcing, a single-step rectified flow–based approximation that aligns training with inference to reduce error accumulation without additional computational cost. Empirical results on the VBench benchmark show improvements over prior methods (FramePack, Magi-1) in background consistency, subject consistency, and motion smoothness, alongside qualitative evidence of better long-term stability and temporal coherence.

**Strengths:**

[1] The paper targets a relevant and challenging problem: maintaining long-term temporal consistency in autoregressive video generation. The proposed MemoryPack module is a reasonable and well-engineered approach to incorporate both short- and long-term dependencies with manageable computational cost. The use of multimodal cues (text and image) as semantic guidance is a thoughtful extension that appears to improve temporal stability in practice.

[2] The Direct Forcing strategy is straightforward but practical. It offers a computationally cheap way to better align training and inference, which is an important consideration for large-scale autoregressive systems.

[3] Experiments are generally well executed, and results show clear though moderate improvements over prior baselines. The writing is clear, figures are helpful, and the paper demonstrates awareness of related work and current limitations.

**Weaknesses:**

[1] Some contributions feel incremental. MemoryPack extends earlier ideas like FramePack rather than introducing a completely new framework. The novelty lies more in integration than in concept.

[2] Direct Forcing, while effective, is close to existing self-forcing or scheduled sampling methods. More theoretical justification or analysis would make the contribution stronger.

[3] Experiments are limited to one-minute videos, leaving open questions about scalability to longer sequences. The human evaluation is useful but lacks methodological detail.

[4] A clearer discussion of failure cases would help readers understand when the method breaks down or struggles to maintain temporal coherence. For instance, Figure 5 highlights artifacts in the 5th to 7th images.

**Questions:**

[1] Could you include experiments with more video diffusion models, e.g., CogVideoX or Wan, etc., under similar training conditions to validate your claim further?

[2] How does the computational and memory footprint of MemoryPack scale with longer videos (e.g., >2 minutes)? Any observed degradation trends?

[3] Could you provide theoretical or empirical evidence explaining why a single-step approximation suffices for stable inference alignment, beyond empirical improvements?

[4] Could the authors report any failure cases where the model loses global semantics or scene structure, to help characterize its remaining limitations?

---

### Official Review · Reviewer_Li7F · 2025-11-01

**Soundness:** 2
**Presentation:** 3
**Contribution:** 3
**Rating:** 4
**Confidence:** 4

**Summary:**

The paper introduces an autoregressive video generation model that enhances long-form video consistency through two key innovations: MemoryPack and Direct Forcing. MemoryPack efficiently models long- and short-term dependencies using text and image guidance, achieving minute-level consistency. Direct Forcing mitigates error accumulation by aligning training and inference in a single step. The model achieves state-of-the-art performance in metrics like Motion Smoothness and Background Consistency, outperforming methods such as FramePack-F1 and Magi-1.

**Strengths:**

- **MemoryPack for Long-Form Consistency**: MemoryPack effectively models both short- and long-term dependencies using text and image guidance, achieving minute-level temporal consistency with linear complexity.
- **Direct Forcing for Training-Inference Alignment**: Direct Forcing mitigates training-inference mismatch through a single-step approximation, reducing error accumulation and improving long-term stability.
- **State-of-the-Art Performance with Visual Validation**: The method achieves state-of-the-art results on key metrics like Motion Smoothness and Background Consistency, outperforming existing models. The paper also provides visualizations and video demos, increasing transparency and allowing for direct assessment of the method's effectiveness.

**Weaknesses:**

- **Missing key references on context memory.** The work overlooks recent advances in memory-based video generation, such as VMem [1], WorldMem [2], and Context as Memory [3], which directly address historical context modeling and retrieval. These should be discussed to properly position the paper.
- **Insufficient evidence for retrieval mechanism.** The claim that MemoryPack retrieves useful historical context lacks strong experimental support. Following the methodology of [1–3], the authors should revisit specific frames and compute frame-level similarity metrics to validate effective retrieval. Otherwise, the paper should clarify that the mechanism primarily mitigates error accumulation and ensures temporal consistency through compression, rather than performing genuine context retrieval—otherwise, it risks misleading the community.

[1] (ICCV 2025) VMem: Consistent Interactive Video Scene Generation with Surfel-Indexed View Memory

[2] (NeurIPS 2025) WORLDMEM: Long-term Consistent World Simulation with Memory

[3] (SIGGRAPH Asia 2025) Context as Memory: Scene-Consistent Interactive Long Video Generation with Memory Retrieval

**Questions:**

- See weaknesses.
- If the authors provide a more thorough analysis of the weaknesses, I would be happy to raise my score.

---

### Official Review · Reviewer_oraA · 2025-11-03

**Soundness:** 2
**Presentation:** 2
**Contribution:** 1
**Rating:** 2
**Confidence:** 5

**Summary:**

This paper proposes two technical modules to generate long-form videos – MemoryPack and Direct Forcing. MemoryPack extends the existing FramePack by adding SemanticPack which preserves long-term semantic information. SemanticPack predicts the future state based on the current frame segment and the previous state with the initial state starting from the text prompt and input image frame for text-to-video generation. Direct Forcing is the post-training strategy to train the model to predict the one-step estimation of the noised input to mitigate the bias of teacher-forcing training, which uses real data samples. Experimental results show incorporating SemanticPack and Direct Forcing improves long video generation quality, enhancing the generated frame consistency over long temporal horizons.

**Strengths:**

S1. This paper tackles the timely and important problem of long video generation. Generating minute-scale videos hold the potential to unblock much broader impacts of real-world applications.

S2. The idea introducing a memory module into long video generation makes sense to jointly maintain long- and short-term consistency. Experimental results also demonstrate the potential of the proposed approach to improve long video generation compared to the baseline.

**Weaknesses:**

W1. Inefficient technical contributions and rationales. The proposed MemoryPack is a tailored and incremental extension to FramePack-F1 for semantic consistency over temporal frames. In addition, there is lack of rationale to claim why MemoryPack is an optimal design to leverage a memory-based module to effectively utilize a long context of inputs. Direct Forcing also lacks enough verification, considering it cannot fundamentally remove the training-inference gap of autoregressive generation.

W2. Lack of experiments and detailed analysis. Although some experimental results are provided to demonstrate the efficacy and effectiveness compared with its baseline, there is no analysis to understand how the proposed module works in detail and compare with other previous studies.

W3. The paper writing should be improved and elaborated for better understanding and reproducibility. Please refer to the questions below.

**Questions:**

In addition to the weaknesses above, I think resolving my concerns below can significantly help improve this paper.

Q1. How does SemanticPack work in detail? Specifically, there are two submodules – Memorize and Squeeze – but their implementation details are insufficient. Line 166 states that Memorize “applies self-attention within windows of historical segments” but Eq. (1) shows that the input of Memorize is only the current frame segment, not historical ones. I also wonder whether the self-attention input or memory state includes any positional bias. In addition, why is the module called “Memorize” for self-attention if there is no external memory that stores information about each frame segment over time?

Q2. While the Squeeze module acts as a recurrent controller, why does it use a cross-attention layer instead of recurrent network variants? Is there any computational or performance advantage in using a cross-attention layer over recurrent networks? Although it employs a cross-attention layer, why is the receptive field of Squeeze restricted to the previous states similar to a recurrent model?

Q3. How does the proposed method determine the memory size? Does the memory size affect performance depending on the duration of the generated videos?

Q4. Is text information mandatory for the MemoryPack module? The text prompt is used for every generation in the MMDiT module. Although zero-MemoryPack performs worse than MemoryPack, using the first frame segment alone might be sufficient.

Q5. Why does the paper adopt two-stage training for Direct Forcing? Could it be integrated into conventional training by randomly applying Direct Forcing with some probability during optimization?

Q6. Could you provide a more detailed analysis by video duration? I believe performance may vary across different generated durations (e.g., 15s, 30s, 60s), which would be valuable to better understand the effect of the memory module.

Q7. How exactly does Direct Forcing reduce the training–inference gap? The gap seems to persist, since the one-step estimation of the flow model is not equivalent to the generated frame during inference. Moreover, when the noise level is high, the one-step estimation may deviate even further from inference samples, potentially increasing the gap. Would a multi-step estimation improve performance? Exploring this variant could clarify the mechanism and effectiveness of Direct Forcing.

Q8. Regarding the experimental results, the reported improvements appear incremental. The gain in ELO score is very small (1568 vs 1537), which could easily reverse due to baseline variance or participant bias. Table 2 shows that aesthetic quality and background consistency drop significantly, even though error accumulation decreases. Conversely, image quality improves while its error accumulation worsens. These inconsistencies make it unclear whether the proposed method yields a significant overall improvement. Additionally, when Direct Forcing is added, the score changes remain inconsistent, making it difficult to identify a clear advantage from the method.

---

### Note · Authors · 2025-11-20

**Comment:**

Dear Editor and Reviewers,

After carefully reviewing the comments received, we have decided to withdraw our submission, “Autoregressive Video Generation with Learnable Memory and Consistent Decoding.”

The reviewers have correctly identified several key aspects of our work that necessitate further development. We believe that fully integrating these insights is essential for the integrity of our research. Consequently, we feel it is best to retract the paper at this stage to allow us the time needed to conduct additional experiments and significantly strengthen the core methodology.

We are grateful for the expert insights shared during this process, which will undoubtedly shape the next iteration of this project.

**Withdrawal Confirmation:**

I have read and agree with the venue's withdrawal policy on behalf of myself and my co-authors.